# Response Surface Modeling and Optimization of the Extraction of Phenolic Antioxidants from Olive Mill Pomace

**DOI:** 10.3390/molecules27238620

**Published:** 2022-12-06

**Authors:** Filipa Paulo, Loleny Tavares, Lúcia Santos

**Affiliations:** 1LEPABE—Laboratory for Process Engineering, Environment, Biotechnology and Energy, Faculty of Engineering, University of Porto, Rua Dr. Roberto Frias, 4200-465 Porto, Portugal; 2ESAN—School of Design, Management and Production Technologies Northern Aveiro, University of Aveiro, Estrada do Cercal 449, Oliveira de Azeméis, 3720-509 Santiago de Riba-Ul, Portugal; 3ALiCE—Associate Laboratory in Chemical Engineering, Faculty of Engineering, University of Porto, Rua Dr. Roberto Frias, 4200-465 Porto, Portugal

**Keywords:** natural extracts, agroindustrial by-product, bioactive compounds, solid-liquid extraction, tyrosol, hydroxytyrosol

## Abstract

Bioactive compounds from olive mill pomace (OMP) were extracted through a two-step solid-liquid extraction procedure considering four factors at five levels of a central composite rotatable response surface design. The influence of the process variables time of the primary extraction (2.0–4.0 h), solvent-to-sample ratio during the primary extraction (5.0–10.0 mL/g), time of the secondary extraction (1.0–2.0 h), and the solvent-to-sample ratio during the secondary extraction (3.0–5.0 mL/g) were examined. The content of bioactive compounds was determined spectrophotometrically, and the individual phenolic compounds were evaluated by reserved-phase high-performance liquid chromatography (RP-HPLC). The Derringer’s function was used to optimize the extraction process, and the best conditions were found to be 3.2 h for the primary extraction, 10.0 mL/g for the solvent-to-sample ratio and 1.3 h for the secondary extraction associated with a solvent-to-sample ratio of 3.0 mL/g, obtaining a total phenolic content of 50.0 (expressed as mg gallic acid equivalents (GAE)/g dry weight (dw). The response surface methodology proved to be a great alternative for reducing the number of tests, allowing the optimization of the extraction of phenolic antioxidants from OMP with a reduced number of experiments, promoting reductions in cost and analysis time.

## 1. Introduction

Epidemiological studies have shown that the incidence of certain cancers, such as breast and colon cancers, as well as coronary heart disease, is lower in the countries around the Mediterranean basin when compared with the northern European countries [1]. This may be related to the safer and more protective dietary patterns in the southern countries where olive oil is the primary source of fat [2,3]. Both in vitro and animal studies suggest that the high concentration of phenolic antioxidants in extra virgin olive oil contributes widely to the healthy pattern of the Mediterranean diet [4]. Nevertheless, the majority of phenolic compounds—around 98%—are retained in olive oil by-products during olive oil processing [5,6]. From this perspective, olive mill pomace (OMP), generated in large quantities during olive oil processing, may be considered a powerful source of natural phenolic antioxidants, such as tyrosol, hydroxytyrosol, and oleuropein [2,7]. Furthermore, there is a growing interest in replacing synthetic food antioxidants with natural ones, which has stimulated research on vegetable sources and raw material screening for the identification of new antioxidants. Therefore, the incorporation of antioxidants in food matrices is required to retain the original color, odor, and flavor to avoid the damage of other vital macromolecules (e.g., vitamins) [2,8]. Natural preservatives and antioxidants include different substances and extracts that can be obtained from a wide variety of plants, grains and fruits [9].

The solvent extraction approach is still the most widely described technique for the extraction of phenolic antioxidants from OMP even though it is generically classified as a time- and solvent-consuming method [10,11]. Even so, the non-requirement of specific and cutting-edge technological equipment, the simplicity of protocols, and the use of mild conditions (e.g., high temperatures would compromise the stability of phenolic antioxidants) encourage the widespread use of this method. Previous findings have reported the influence of independent variables, such as the extraction procedure, the solvent composition, the pH, the temperature, the extraction time, the solid-to-liquid ratio and the inclusion of pre-treatment steps, such as fat removal and acidic and/or alkaline hydrolysis on the solvent extraction of phenolic antioxidants from OMPs [12,13,14,15]. The disadvantages of this traditional extraction method include: the long extraction times, high energy consumption, the need for expensive, high-quality organic solvents, evaporation of the solvents during treatment and the use of these potentially toxic solvents, which then pose problems with disposal and possible loss of functionality [16]. In this sense, there are some environmentally friendly techniques applied for the recovery of polyphenols from food by-products [16]. Therefore, the optimization of the extraction process can decrease these adverse effects, and among the available statistical and mathematical models for the multivariate analysis, the response surface methodology (RSM) is one of the most commonly used [17,18]. The study of one variable while the others remain constant—known as the one-variable-at-time (OVAT) or interchangeably one-factor-at-time (OFAT) methodology—generally leads to non-optimized responses/outputs as the OVAT methodology does not consider the fact that different variables can interact and have an impact on the response and therefore on the optimization process [17,18]. The RSM encloses a set of mathematical and statistical approaches to express relationships between factors and responses [17].

The goal of this study was to investigate the individual and interactive effects of two-step solid-liquid extraction process variables such as the time of the primary extraction (hours), the solvent-to-sample ratio (SSR) during the primary extraction step (mL/g), the time of the secondary extraction (hours) and the SSR during the secondary extraction step (mL/g) on the total phenolic content (TPC) of OMP extracts. It was also intended to optimize the process variables of the solid-liquid extraction procedure, aiming for the maximization of the phenolic content of OMP extracts using a central composite rotatable response surface design associated with the Derringer’s desired function methodology. Additionally, the extract obtained under optimized extraction conditions was characterized regarding its TPC, total antioxidant activity (TAA), and the total amount of specific biophenols (hydroxytyrosol, tyrosol, and oleuropein). The distribution of phenolic compounds and antioxidants among OMP extract fractions was evaluated. Furthermore, the TPC and TAA of the extract obtained under optimized extraction conditions were compared to the TPC and TAA of the extract obtained without considering OMP pre-treatment, evaluating the extent effect of acidic hydrolysis on the extraction of phenolic compounds from OMP samples via a two-step solid-liquid extraction procedure.

## 2. Results and Discussion

A design of 30 two-step solid-liquid extraction batch experiments was performed to study and optimize the individual and combined effects of the selected variables (time of primary extraction, sample-to-solvent ratio during the primary extraction, time of secondary extraction and sample-to-solvent ratio during the secondary extraction) on the total phenolic content of OMP extracts. In Table 1 (described in the section on materials and methods), the experimental design (variables), the experimental/observed values (OV), and the predicted values (PV) for the response are presented, as are the residual error (RE) and the percentage of the error of each experiment on the response

### 2.1. Central Composite Rotatable Design Analysis

The regression equation that describes the dependence of the response (TPC) of the variables (time of the primary extraction, SSR during the primary extraction, time of the secondary extraction, SSR during the secondary extraction) was obtained by fitting the experimental data into alternative models like the (i) the linear model, (ii) the two-factor interaction (2FI) model, (iii) the quadratic model and (iv) the cubic model. For the evaluation of the adequacy of the models, three different tests were conducted: (i) the sequential model sum of squares, (ii) the lack-of-fit test; and (iii) the model summary statistics. The results are presented in Table 2.

Accordingly, for the TPC of OMP extracts, the sequential model sum of squares presented a *p*-value lower than 0.0001 for the quadratic model, and therefore this model is suggested to describe the dependence of the TPC of the process variables. Regarding the results of the lack-of-fit test, the highest lack-of-fit *p*-values were obtained for the quadratic and cubic models (the lack-of-fit should be insignificant and lack-of-fit *p*-values > 0.10 are desirable). Accordingly, the quadratic model is suggested by the lack-of-fit test as the cubic model is aliased (the RSM model chosen was too small to estimate a cubic model). The model summary statistics list other relevant statistics used to compare models. This model focuses on the model, maximizing the adjusted R^2^ and the predicted R^2^. The maximization of both the adjusted R^2^ and the predicted R^2^ was possible using the quadratic model. Moreover, it was observed to be the lowest value for the predicted residual error sum of squares (PRESS) for the quadratic model. The PRESS statistic is a form of cross-validation. Generally, the lowest values of PRESS indicate more robust regression models.

The fit summary for the TPC response indicates that the quadratic model should be selected as for this model was observed the lowest sequential *p*-value (*p*-value < 0.00001), a robust lack-of-fit *p*-value (lack-of-fit *p*-value > 0.1) and reasonable high values for the adjusted and the predicted R^2^. Therefore, the quadratic model incorporating linear, the 2FI and the quadratic terms was chosen to describe the effect of the time of the primary extraction, the SSR during the primary extraction, the time of the secondary extraction, and the SSR during the secondary extraction on the two-step extraction procedure of phenolic antioxidants from OMP.

### 2.2. Statistical Analysis

The statistics analysis of variance (ANOVA) and the multiple regression analysis were considered for the evaluation of the fitness of the quadratic model chosen to the experimental data. The results of these analyses are presented in Table 3. The statistical significance of the regression equation was evaluated, considering models F and *p*-values. The quadratic model presented an F-value of 40.16, which implies that the model is significant. There was only a 0.01% chance that an F-value this larger could occur due to noise. The obtained *p*-value of the quadratic regression model was lower than 0.0001. The quadratic model was highly statistically significant for the TPC. The lack-of-fit test was employed to analyze the fitness of the model. The quadratic regression model presented a lack-of-fit F-value of 2.63 and a lack-of-fit *p*-value of 0.1488, which indicated that the selected model is suitable to predict variations. The obtained lack-of-fit F-value implied that the lack-of-fit is not significant in relation to the pure error. There was only 14.88% chance that a lack-of-fit F-value this large could occur due to noise. The goodness of fit of the model was evaluated considering the statistics (i) determination coefficient ($R^{2}$), (ii) the predicted determination coefficient ($R_{a}^{2}$), (iii) the predicted determination coefficient ($R_{p}^{2}$), (iv) the adequate precision, and (v) the coefficient of variance (CV). It was observed $R^{2}$ and $R_{a}^{2}$ values of 0.974 and 0.950, respectively. According to Yetilmezsoy, et al. [19], if the sample size is not large enough or if there are many terms in the model, the $R_{a}^{2}$ may be significantly smaller than the $R^{2}$. In this study, the difference between $R^{2}$ and $R_{a}^{2}$ values was low (0.024) for the quadratic regression—a high degree of correlation between the experimental and the predicted values was found. The $R_{a}^{2}$ and the $R_{p}^{2}$ values should be reasonably close—a difference of less than 0.2 is desired. The quadratic regression for the TPC presented $R_{a}^{2}$ (0.950) and $R_{p}^{2}$ (0.868) values in reasonable agreement ($R_{a}^{2}-R_{p}^{2}$ = 0.082 < 0.2). The adequate precision, also known as the signal-to-noise ratio, allows the comparison between the range of the predicted values at the design points and the average prediction error. A ratio greater than 4 indicates adequate model discrimination. It was observed an adequate precision of 27.7 highly than 4 for the model adequacy for the TPC, indicating an adequate precision of the evaluated model. The CV reflects the ratio of the standard error of the estimate to the average value of the observed response, establishing the reproducibility of the model. The low CV value (8.4%) observed indicates that the deviations between the experimental and predicted values were reasonably low, and the model demonstrated a high degree of precision, also indicating a satisfactory reliability of the experiments performed. The PRESS value was found to be the lowest (PRESS = 634.0) when compared to the PRESS values obtained for the other regressions (e.g., mean model PRESS = 5150.1, linear model PRESS = 6785.6, 2FI model PRESS = 3827.5). Therefore, it can be concluded that the quadratic model fitted each point in the design better than the other models considered (mean, linear, and 2FI models). A quadratic regression model was selected to establish the relationship between the factors and the response. The model F-value and the associated *p*-values confirmed the model’s significance. Moreover, the $R_{p}^{2}$ and $R_{a}^{2}$ were in reasonable agreement (within 0.2), the CV was remarkably low, and the model exhibited adequate precision at a sufficiently high level (>4). Therefore, it can be concluded that the model chosen afforded a systematic explanation of the relationship between the factors (X_1_: time of primary extraction, X_2_: SSR during the primary extraction, X_3_: time of secondary extraction, and X_4_: SSR during the secondary extraction) and the response (TPC expressed as mg GAE/g dw OMP).

### 2.3. Diagnostics of the Adequacy of the Models

It is crucial to confirm if the fitted model provides enough similarity between the predicted and experimental values. Therefore, in addition to the ANOVA and the multiple regression analysis, the adequacy of the selected model was evaluated through the analysis of the residuals using graphical methods. Normal probability plots are appropriate graphical methods to evaluate the residuals’ normality. Residuals, defined as the difference between the observed values and the predicted response values, are elements of variation that cannot be explained by the fitted model, which is expected to follow a normal distribution. The observed residuals expressed in terms of externally studentized residuals were plotted against the normal probability (%) (Figure 1A). The design point values lay reasonably close on the straight line. Some scatter was found—which is not considered an unusual occurrence. Nevertheless, if a defined pattern is observed (e.g., an ‘S-shaped’ curve), a transformation of the response may provide a better basis for analysis. It was not observed in a defined pattern. This assumption was validated through the analysis of the Box-Cox Plot for Power Transforms (Figure 1B). The best lambda value (λ = 0.683)—found at the minimum point of the curve generated by the natural logarithm of the sum of squares of the residuals—considering a 95% of confidence interval (0.157 ≤ λ ≤ 1.126) included the lambda of 1, and therefore, no power transformation was considered. Therefore, it can be concluded that there was observed a normal distribution of the data for the response, supported by the Box-Cox Plot for Power Transformations (Figure 1B). The diagnostic plot of predicted versus actual design values is presented in Figure 1C. There was observed to be a strong correlation between model predictions and their actual values as the data points lay close to the straight line. The Cook’s distance plot (Figure 1D) displays that all the data points were within the desired limits, and no outliers were found in the design since the Cook’s distance values were below the defined limit (Cook’s distance of 1).

### 2.4. Percentage Contribution of Process Variables

The PC of each process variable for the linear, 2FI, and quadratic effects was evaluated (Figure 2). The PCs were calculated considering the coded regression coefficients from the ANOVA. The 2FI between the SSR during the primary extraction and the time of the secondary extraction (X_2_X_3_) (21.0%) followed by the 2FI between the time of the secondary extraction and the SSR during the secondary extraction (X_3_X_4_) (19.0%) exhibited the highest effect on the extraction of phenolic antioxidants from OMP. On the opposite, the 2FI between the time of the primary extraction and the SSR during the primary extraction (X_1_X_2_) (0.01%) followed by the SSR during the secondary extraction (X_4_) (0.08%) showed to excrete the lowest impact on the extraction procedure (Figure 2).

### 2.5. Assessment of the Influence of Process Variables

A CCRD considering four factors coded at five levels was selected to examine the influence of the time of the primary extraction (X_1_, hours), the SSR during the primary extraction (X_2_, mL/g), the time of the secondary extraction (X_3_, hours) and the SSR during the secondary extraction (X_4_, mL/g) on the two-step solid-liquid extraction of phenolic antioxidants. One-variable plots and three-dimensional (3D) response surface plots were plotted to evaluate the effect of each factor considered individually and the interaction among the independent variables, respectively, on the TPC of OMP extracts. The 3D response surface plots were obtained by considering two factors at a constant level while the other two factors were changed.

#### 2.5.1. Assessment of the Influence of Individual Process Variables on the Total Phenolic Content of Olive Mill Pomace Extracts

The effect of each individual variable on the extraction of phenolic compounds, estimated through the total phenolic content of OMP extracts, was evaluated through the analysis of one-variable plots (Figure 3). The time of the primary extraction influenced non-linearly the extraction of phenolic antioxidants from OMP extracts (Figure 3A). The highest phenolic content was observed in the experiments conducted considering the lowest and the highest values of the process variable time of the primary extraction. The lowest phenolic content was observed for experiments in which the median values of the variable X_1_. A study regarding the optimization of a solid-liquid extraction procedure was performed by Alu’datt, Alli, Ereifej, Alhamad, Al-Tawaha and Rababah [12]. The authors considered alternative conditions to the ones exposed in the present research during an OVAT optimization, such as the solvent, the number of extraction steps, temperature, and others. Contrarily to the results obtained in the present study, the authors inferred that increasing the time of extraction lead to an increase in the total phenolic content of the OMP extracts obtained. Even though a direct comparison should be performed carefully as many variables were different between the two studies, it can be stated that, probably, considering the extraction conditions proposed by the authors, the required extraction time to observe a decrease in the phenolic content of OMP extracts was not reached. Moreover, it is expected a decrease in the total phenolic content of OMP extracts with the extraction time due to degradation processes that may occur, which were not observed by Alu’datt, Alli, Ereifej, Alhamad, Al-Tawaha and Rababah [12]. The results of the present study are supported by the results obtained by Rubio-Senent, et al. [20]. The authors performed the extraction of phenolic compounds through a hydrothermal treatment and observed that the amount of phenolic compounds extracted did not increase linearly with the increase in the extraction time. A similar trend to the process variable X_1_, was observed with the process variable SSR during the primary extraction (X_2_). It was observed that the highest phenolic content in the experiments was associated with the lowest and highest values of the SSR during the primary extraction. The minimum phenolic content was observed in the experiments, which were considered the median values of SSR (X_2_) (Figure 3B). The phenolic and antioxidant potential of olive mill wastes was evaluated by Lafka, et al. [21]. The authors observed an increase in the TPC of olive mill waste extracts with the increase in the SSR value. Nevertheless, the authors demonstrated that for SSR values higher than 5 v/w, it was almost impossible to extract more phenolic antioxidants (a plateau on the total phenolic content of extracts versus the SSR value was reached). In contrast, the results of this study demonstrated that a plateau has not been reached: an increase in the TPC was observed from the median to the highest SSR values. The addition of more extraction solvent favors the non-saturation of the extraction solvent, and therefore more phenolic compounds can be extracted from OMP into the extraction solvent. The time of the secondary extraction influenced the extraction of phenolic antioxidants in an opposite trend to the process variable X_1_ (time of the primary extraction). In the case of the dependence of the total phenolic content of OMP extracts on the time of the secondary extraction (X_3_), it was observed that the median values of the process variable X_3_ were maximized. The lowest phenolic content was observed in experiments in which the lowest and highest values were considered for the time of the secondary extraction (Figure 3C). Contrasting with the non-linear pattern dependence of the TPC of OMPs from the X_1_, X_2,_ and X_3_ variables, the X_4_ variable—SSR during the secondary extraction—did not affect largely the efficient extraction of phenolic antioxidants when considered individually. It was observed a non-dependence of the single factor SSR during the secondary extraction corresponding to a slope of the line equal to zero (Figure 3D).

#### 2.5.2. Assessment of the Influence of Combined Process Variables on the Total Phenolic Content of Olive Mill Pomace Extracts

Experiences were conducted to observe the effect of the time and the SSR during both the primary and secondary extractions on the recovery of phenolic antioxidants estimated through the TPC of OMP extracts. It was observed that for the lowest SSR during the primary extraction (5.0 mL of ethanol/g of OMP), the time of primary extraction had no influence on the TPC. Nevertheless, when the SSR was increased, it was also observed that the TPC increased along with the time of the primary extraction. For a SSR of 10.0 mL/g and 2 h for the primary extraction, the extracts exhibited a TPC of 34.4 mg GAE/g of OMP. However, when the extraction time was doubled while keeping the SSR constant (10.0 mL/g), the TPC increased (Figure 4A). When the 3D response surface plot was plotted, exhibiting the effects on the TPC of the contributions of the time of the primary extraction and the time of the secondary extraction, a compromise was observed between the times of each extraction step. The longest secondary extraction time did not appear to have a positive effect on the extraction of phenolic compounds. However, the median times for the first extraction stage (around 3.0 to 3.5 h) associated with reduced times during the second extraction stage, seemed to favor the extraction of phenolic compounds (Figure 4B). The relationship between the time of the primary extraction and the SSR during the secondary extraction was challenged to infer. It was observed that from the median to low SSRs, the recovery of phenolic compounds increased, especially for the lowest and highest values for the time of the primary extraction. From the median to higher SSR for the secondary extraction, the recovery of phenolic compounds was almost independent of the time of the primary extraction (Figure 4C). In Figure 4D is depicted the influence of the SSR during the primary extraction and the time of the secondary extraction. It was observed that the combination of low values for these two variables did not favor the extraction of phenolic compounds. The TPC of OMP extracts was increased with the simultaneous increase of the SSR during the primary extraction and the decrease in time of the secondary extraction. When the SSRs were plotted to infer their influence on the TPC, it was observed that the maximum total phenolic content was achieved when considering the highest SSR value during the primary extraction combined with the lowest SSR value during the secondary extraction (Figure 4E). The relationship between time and SSR value during secondary extraction was not linear. The combination of the low time of extraction and high SSR value during the secondary extraction negatively affected the extraction of phenolic compounds. The highest recovery of phenolic compounds was observed at low SSR values combined with median times for the secondary extraction (Figure 4F). No data was found in the literature for the direct comparison of the obtained results of the present study.

### 2.6. Optimization of Process Variables and Validation of the Optimized Conditions

The Derringer’s desired function methodology was applied to optimize the extraction conditions with the purpose of maximizing the extraction of phenolic antioxidants estimated through the TPC of OMP extracts. Accordingly, the optimal extraction conditions were found to be a time of 3.2 h for the primary extraction (X_1_), SSR during the primary extraction of 10.0 mL/g (X2), a time of 1.3 h for the secondary extraction (X3), and SSR during the secondary extraction of 3.0 mL/g (X_4_). Considering the optimized conditions, the predicted TPC was 50.0 mg GAE/g dw OMP, associated with a desirability value of 0.717. A desirability ramp was generated from the optimal points through a numerical optimization approach (Figure 5).

The accuracy of the prediction of the optimum response values, which indicate the suitability of the optimized extraction conditions, was evaluated by comparing the obtained experimental results with the post-analysis of the optimization procedure. As presented in Table 4, the observed TPC was 50.5 ± 1.5 (n = 3) which is within the predicted interval (95% confidence interval) of the model (lowest predicted value by the model = 45.2 < obtained value = 50.5 < highest predicted value by the model < 54.7). Therefore, it can be concluded that the model accurately predicted the optimum TPC considering optimized extraction conditions of phenolic antioxidants from OMP samples.

### 2.7. Characterization of the Optimized Olive Mill Pomace Extracts

The extract (n = 3, where n represents the number of responses in the study) obtained considering the optimized extraction conditions (time of the primary extraction of 3.2 h; SSR during the primary extraction of 10 mL/g; time of the secondary extraction of 1.3 h; SSR during the secondary extraction of 3.0 mL) was characterized regarding its TAA and TPC. Moreover, the contribution of each fraction—phenolic antioxidants extracted from the fat removal step, phenolic antioxidants recovered during the primary extraction, and the phenolic antioxidants extracted during the secondary extraction—to the TAA and TPC was evaluated. Böhmer-Maas, et al. [22] studied the optimization of the extraction of phenolic compounds from olive pomace using response surface methodology. According to the authors, the conditions that promoted the highest TPC in an extract were using 40% methanol, 70 °C and 180 min. The highest AA was in the extract obtained with 40% methanol, 45 °C and 180 min. The phenolic profile and the quantification of specific phenolic antioxidants were assessed by RP-HPLC analysis of the final extract. Furthermore, the effect of the application of an acidic hydrolysis step prior to the solid-liquid extraction of phenolic antioxidants from OMP samples was evaluated generically for antioxidants and specifically for phenolic compounds, considering the total, the bounded and the free phenolic antioxidant distributions in hydrolyzed and non-hydrolyzed extracted samples.

#### 2.7.1. Contribution of Alternative Extract Fractions on the Total Phenolic Content and the Total Antioxidant Activity

The contribution of each fraction (EPAS-FR—Extracted Phenolic Antioxidants during the Fat Removal Procedure; EPAS-P1E—Extracted Phenolic Antioxidants during the primary extraction; EPAS-P2E—Extracted Phenolic Antioxidants from the secondary extraction procedure) on the (i) the total phenolic content and (ii) the total antioxidant activity was evaluated (Figure 6). The amount of phenolic compounds extracted from the EPAS-FR was 0.6 ± 0.1 mg GAE/g dw OMP corresponding to only 1.2 ± 0.2% of the total phenolic content of OMP extracts obtained considering optimized extraction conditions. There were not observed significant differences (*p* = 0.83 > 0.05) on the TPC among the EPAS-P1E and EPAS-P2E fractions. The TPC of the fractions EPAS-P1E and EPAS-P2E were 26.1 ± 1.5 mg GAE/g dw OMP (percentage proportion of 51.7 ± 3.0% of the TPC) and 23.8 ± 0.7 mg GAE/g dw OMP (percentage proportion of 47.1 ± 1.4% of the TPC), respectively. The total amount of polar phenolic compounds (fractions EPAS-P1E and EPAS-P2E) was 49.9 ± 1.3 mg GAE/g dw OMP, corresponding to a proportion of 98.8 ± 2.3% of the TPC. The high amount of phenolic compounds present in EPAS-P1E and EPAS-P2E fractions and the low amount of phenolic compounds recovered from the fat removal step (EPAS-FR fraction) reveals the majority of the extracted phenolic compounds from OMP samples are from hydrophilic nature (aromatic rings with attached hydroxyl groups in their structures with strong polar character). Furthermore, the present results demonstrate that ethanol can be safely used as an extraction solvent of phenolic compounds from OMP samples, unraveling new perspectives for the replacement of toxic solvents commonly used in solid-liquid extraction of phenolic compounds from OMP samples (e.g., methanol). A similar pattern was found for the TAA distribution between the fractions EPAS-FR, EPAS-P1E, and EPAS-P2E. The fraction EPAS-FR exhibited only 2.4 ± 0.5% of TAA, corresponding to a percentage contribution to the TAA of 3.0 ± 0.6%. There was no statistically significant difference in the TAA between the fractions EPAS-P1E and EPAS-P2E (*p* = 0.78 > 0.05). The TAA of EPAS-P1E and EPAS-P2E fractions were 45.7 ± 4.1% (percentage contribution to the TAA of 57.8 ± 5.2%) and 30.9 ± 3.6% (percentage contribution to the TAA of 39.1 ± 4.6%), respectively. The two-step extraction (EPAS-P1E and EPAS-P2E fractions) using ethanol as the extraction solvent (combined TAA of EPAS-P1E and EPAS-P2E of 76.7 ± 7.7%) contributed to 97.0 ± 9.7% for the TAA of the final extract. These results reinforce the hypothesis that the majority of the antioxidants extracted were phenolic compounds. From the statistical analysis, as it was not observed that there were significant differences in the TPC and TAA of both EPAS-P1E and EPAS-P2E fractions, it can be confirmed that the secondary extraction is essential for the recovery of phenolic antioxidants from OMP samples. In this context, a multi-step extraction should be considered when it is intended for the recovery of phenolic antioxidants from OMP samples for their further valorization. To the authors’ best knowledge, there was a lack of published data suitable for a direct comparison regarding the obtained results.

#### 2.7.2. RP-HPLC Analysis of Phenolic Antioxidants

The RP-HPLC analysis of the present study was focused on the identification, detection, and quantification of two OMP phenyl alcohols—hydroxytyrosol and tyrosol—and a glycosylated seco-iridoid compound—oleuropein. Oleuropein is the major biophenol present in olives and olive leaves [23]. Oleuropein is a heterosidic ester of elenolic acid and 3,4-dihydroxyphenylethanol, whose hydrolysis yields hydroxytyrosol—one of the most potent phenyl alcohols present in olive oil, olive products, and by-products—and elenolic acid glucoside [24]. Nevertheless, oleuropein is often hydrolyzed to hydroxytyrosol during olive oil processing as a high amount of water is required during the olive oil extraction process [25]. Hydroxytyrosol is considered one of the most potent antioxidants present in olive oil and olive oil by-products [23,26]. In contrast, tyrosol can be found in olive oil, olive oil products, and olive oil by-products as a free phenol or in its oleoside form (tyrosol glucoside) [27]. As complex chemical transformations occur during olive fruit maturation and olive oil processing, the antioxidants hydroxytyrosol, tyrosol, and oleuropein may be found in OMP samples in their free forms, along with several other analogs [28]. Moreover, the phenolic profile of OMP extracts is widely affected by the olive fruit variety, climatic conditions, agronomic practices, OMP storage and handling, phenolic compounds extraction procedures, among others [29]. The RP-HPLC identification of the compounds hydroxytyrosol, tyrosol, and oleuropein was performed by comparison with the respective analytical standards. The hydroxytyrosol and tyrosol concentrations on the OMP extract obtained under optimized extraction conditions were 2870.2 ± 1.2 mg/kg dw OMP and 1622.1 ± 2.2 mg/kg dw OMP, respectively. The obtained concentrations for oleuropein were below of both limit of detection (LOD) and limit of quantification (LOQ); therefore, it can be concluded that oleuropein was converted into hydroxytyrosol and/or other analogs during phenolic antioxidants extraction. Comparable results were described by a previous investigation of Japón-Luján and Luque de Castro [30]. The authors described the extraction of hydroxytyrosol, tyrosol, and other olive biophenols using static-dynamic superheated liquids. Similarly, in the present study, a multivariate methodology was employed to optimize the extraction conditions. In that investigation, the authors reported it was possible to achieve a total amount of hydroxytyrosol and tyrosol of 2872 mg/kg dw OMP and 1565 mg/kg dw OMP, respectively, in the extract obtained considering the optimized extraction conditions. Moreover, a higher amount of hydroxytyrosol was observed than tyrosol, which is in line with the majority of the published investigations on the extraction of phenolic compounds from OMP [14,20,31].

#### 2.7.3. Effect of Acidic Hydrolysis Pre-Treatment on the Total Phenolic Content and the Total Antioxidant Activity

The effect of the application of an acidic hydrolysis step prior to the solid-liquid extraction of phenolic antioxidants from OMP samples was evaluated generically for antioxidants and specifically for phenolic compounds, considering the total, the bounded and the free phenolic antioxidants distributions in both hydrolyzed extracted samples (HS) and non-hydrolyzed extracted samples (NHS). Considering both HS and NHS, in each type of extract, it was observed a significant difference (*p* < 0.05) in the distribution of bounded and free phenolic compounds. Similarly, it was also noticed that there were statistically significant differences (*p* < 0.05) between bounded and free phenolic compounds concentrations between HS and NHS. The highest phenolic content was observed in hydrolyzed samples (50.5 ± 1.5 mg GAE/g dw OMP) (Table 5). The TPC of extracts obtained considering the optimized extraction conditions, and not considering the HCl hydrolysis step, exhibited only 43.2 ± 1.4% of the phenolic content of the samples obtained under optimized extraction conditions (21.7 ± 0.7 mg GAE/g dw OMP). This suggests that the acidic hydrolysis step is crucial for the recovery of phenolic compounds. The obtained results are in agreement with the ones obtained by Lafka, Lazou, Sinanoglou and Lazos [21] when the authors were describing the phenolic and antioxidant potential of OMP samples submitted to a solid-liquid extraction. The authors submitted the OMP samples to hydrolysis at different pH values, verifying a higher recovery of phenolic compounds at the lowest pH selected (pH 2). Likely, the acidic hydrolysis had promoted the solubilization of phenolic antioxidants in the extraction solvent through two alternative mechanisms: (i) the detachment of phenolic compounds linked to OMP cell walls by ester or glycoside bonds and through the (ii) promoting of OMP cell wall disintegration. In the case (i) of phenolic compounds detaching from OMP cell walls, the acidic environment promoted the cleavage of the bonds in which cell wall phenolic antioxidants were linked (ester and glycosidic bonds), which has promoted the solubilization of cell wall phenolic antioxidants in the extraction solvent. Additionally, the acidic hydrolysis (ii) promoted OMP cell wall disintegration, enabling the diffusion and solubilization of these compounds in the extraction solvent, increasing their stability. In the case of extracts obtained considering the optimized extraction conditions, the contribution of bounded and free phenolic compounds to the TPC of the extracts was 26.9 ± 0.5% (13.6 ± 1.1 mg GAE/g dw OMP) and 73.1 ± 0.8% (36.9 ± 0.4 mg GAE/g dw OMP), respectively (Table 5). It was observed that the amount of bounded phenolic compounds extracted increased when the 12-hr hydrolysis was considered (bounded phenolic compounds in NHS: 2.0 ± 0.1 mg GAE/g dw OMP; bounded phenolic compounds in HS: 13.6 ± 1.1 mg GAE/g dw OMP), leading to an increase of bounded phenolic compounds contribution from 9.2 ± 0.5%—in the case of non-hydrolyzed samples—to 26.9 ± 2.2%—in the case of hydrolyzed samples. Therefore, it can be concluded that the acidic hydrolysis favored the recovery of both bounded and free phenolic compounds. The highest recovery of phenolic compounds when the acidic hydrolysis was considered can be explained considering (i) the stability of free phenolic antioxidants in the extraction solvent, (ii) the detachment of OMP cell wall bonded phenolic antioxidants, and (iii) the disruption of OMP cell walls and the consequent diffusion and solubilization of phenolic antioxidants in the extraction solvent.

A similar pattern was observed regarding the TAA of HS and NHS. In both types of extracts (HS and NHS), it was observed significant differences (*p* < 0.05) on the TAA between the free and the bounded fractions. Similarly, it was verified significant differences (*p* < 0.05) on the TAA between the HS and NHS. The HS exhibited higher TAA then NHS (TAA of HS: 79.1 ± 7.9%; TAA of NHS: 53.5 ± 3.9%) (Table 5). The TAA of NHS corresponded to 67.6 ± 4.9% of the TAA of HS. The percentage contributions of the bounded and the free extract fractions were, respectively, 36.8 ± 4.9% and 63.1 ± 5.9% in the case of HS and 35.1 ± 7.3% and 64.9 ± 7.1% in the case of NHS. Roughly, the percentage contributions of each fraction (free and bounded antioxidants) were similar in both HS and NHS (about 60% of proportional contribution of free antioxidants and 40% of proportional contribution of bounded antioxidants) even though, overall, HS samples present higher TAA. The hydrolysis prior to the solid-liquid extraction promotes the dissolution of bounded phenolic compounds in the extraction solvent (proportion of bounded phenolic compounds extracted from HS was circa 70%; the proportion of free phenolic compounds extracted from NHS was around 10% of the total amount of extracted phenolic compounds); nevertheless, as the proportions of extracted antioxidants from free and bounded fractions were respectively around 60% and 40% for both HS and NHS, it can be hypothesized that the TAA of extracts submitted to hydrolysis was positively affected by the dissolution of bounded phenolic compounds. To the authors’ best knowledge, this is the first time that is reported the extraction of phenolic antioxidants from OMP using two-step solid-liquid extraction, including pre-treatment steps of the samples as hydrolysis and fat removal procedures. Therefore, there is a lack of published data suitable for a direct comparison regarding the obtained results. Nevertheless, some bibliographic data regarding the total phenolic content of extracts obtained from OMPs considering alternative extraction conditions were found in the literature, allowing an outward comparative analysis. The authors Obied, et al. [32] performed the extraction of phenolic antioxidants from Frantoio olive fruit (90% black skin coloration), processed in a two-phase olive oil extraction system in WaggaWagga, Australia. The extraction was performed with a solvent mixture of methanol and water at a ratio of 80/20 *v*/*v*, at pH 2; it was also described as a lipid removal step with *n*-hexane. The authors reported that the phenolic content of the obtained extracts was 17.7 ± 0.9 mg GAE/g dw OMP. The results obtained in the present study allowed a higher recovery of phenolic antioxidants, considering the optimized extraction conditions (50.5 ± 1.5 mg GAE/g dw OMP). Nevertheless, it should be taken into consideration that the total phenolic content of OMP extracts is widely dependent on the geographic origin of olives, cultivar factors (e.g., olives type, climatic conditions), phenolic antioxidant extraction techniques, among others [24].

## 3. Materials and Methods

The olive mill pomace (OMP) samples were obtained from a continuous two-phase centrifugation system by a local olive oil mill in Portugal (Vilas Boas, Vila Flor, Bragança, 41359822, -7123743). Folin-Ciocalteu reagent, gallic acid standard, Trolox standard, 2,2-diphenyl-1-picrylhydrazyl (DPPH), sodium carbonate anhydrous were purchased from Sigma Aldrich Chemical (St. Louis, MO, USA). Hydroxytyrosol, tyrosol and oleuropein were also obtained from Sigma Aldrich Chemical (St. Louis, MO, USA). Solvents such as hexane, ethanol, ethyl acetate, acetonitrile and methanol were obtained from VWR International (Fontenay-sous-Bois, France). Additionally, hydrochloric acid solution at 37% *v*/*v* and acetic acid standardized solution 0.1 N were also obtained from VWR International (Fontenay-sous-Bois, France). All the reagents were either chromatographic or analytical grade and used as received.

### 3.1. Extraction of Phenolic Antioxidants

The olive mill pomace (OMP) samples were stored at −22 °C before analysis. The samples were freeze-dried for 72 h on a benchtop freeze-dryer (SP Scientific, Warminster, PA, USA) (moisture content of 71.0 ± 5.5 % *w*/*w*) and ground on an electric mill (Qilive Q5321 Grinder) to an average particle size of 142.2 ± 9.6 μm. The mean particle size of OMP was evaluated by laser granulometry technique using a Coulter Counter-LS 230 Particle Size Analyzer (Miami, FL, USA) equipment. The extractions of bioactive compounds were performed in three stages: (i) sample pretreatment and bonded phenolic compounds extraction, (ii) two-step solid-liquid extraction of free phenolic compounds, and (iii) liquid-liquid extraction of these valuable bioactive compounds, as presented in the following sections. Grounded freeze-dried OMP samples (1 g/sample) were submitted to pre-treatment procedures prior to the extraction of phenolic antioxidants, such as (i) acidic hydrolysis and (ii) fat removal, according to the method described by Paulo, Tavares and Santos [7]. The (i) acidic hydrolysis promotes the breakage of both glycosidic and ester bonds [17], and (ii) the fat removal procedure was considered, as lipids, due to their lipophilic nature, are interfering agents in obtaining hydroalcoholic extracts [7]. Samples containing the phenolic compounds recovered from the fat removal procedure were termed as samples containing extracted phenolic compounds from fat removal procedure (SEPC-FR) were stored in amber flasks under dark conditions at 20 °C before being admixed with the final filtrates. Subsequently to the fat removal, phenolic antioxidants present in the acidified and defatted OMP samples were extracted in a two-step procedure using ethanol as the extraction solvent. The alternative process conditions tested are presented in Table 1. The hydrolyzed and defatted OMP samples were transferred to 50 mL Erlenmeyer flasks prior to the primary extractions. During the primary extractions, pre-defined volumes of the extraction solvent (corresponding to the SSR values for the primary extraction; Table 1) were added to the respective OMP sample. The obtained solutions were continuously shaken in an orbital shaker at 20 °C for a pre-defined interval of time. Afterward, the solid and liquid phases were separated through centrifugation (2670 G, 15 min). The supernatants recovered from the primary extractions (P1E) were filtered under vacuum using Whatman™ nylon membrane filter and stored in an amber flask under dark conditions at 20 °C. The phenolic antioxidants extracted from the primary extractions were named as extracted phenolic antioxidants samples from primary extraction (EPAS-P1E) (Appendix A. The pellet samples were reconstituted in a pre-defined volume of the extraction solvent and transferred to 50 mL Erlenmeyer flasks for the subsequent secondary extraction processes. The secondary extraction processes were performed similarly to the described processes for the extraction of phenolic antioxidants during the primary extractions, and the phenolic antioxidants extracted from the secondary extraction’s procedures were designated as extracted phenolic antioxidants samples from secondary extraction (EPAS-P2E). For each extraction, the EPAS-FR, EPAS-P1E, and EPAS-P2E were combined, and the final filtrates were submitted to a liquid-liquid extraction using ethyl acetate as the extraction solvent. A volume of ethyl acetate corresponding to three times the volume of each final filtrate samples was admixed with the final filtrate sample, vigorously shaken, vortexed for 5 min, ultrasonicated for 15 min and centrifuged (2670 G, 15 min). Afterward, the phases were separated, and for each sample, ethyl acetate was removed at 50 °C using a rotary evaporator (BUCHI R-210, Buchi Laboratotiums Tchnik AG, Flawil, Switzerland). Solvent traces were removed by a gentle nitrogen stream. The final extracts were stored at −22 °C before prior analyses. A schematic representation of the extraction procedure employed is presented in Appendix A.

### 3.2. Characterization of the Extracts

The obtained extracts were allowed to come to room temperature (20 ± 2 °C) before use. Afterward, the extracts were reconstituted in 3 mL of UPW and stored light protected before prior analyses.

#### 3.2.1. Determination of the Total Phenolic Content and Antioxidant Activity

The TPC of OMP extracts was determined according to the Folin-Ciocalteu procedure, as described by Singleton, et al. [33], and the total antioxidant activity (TAA) of OMP extracts was estimated through the determination of the free radical-scavenging ability (RSA) of extracts using the stable 2,2-diphenyl-2-picrylhydrazyl radical (DPPH^•^) [34].

For the TPC, the calibration curve obtained was based on the mean of three independent calibration curves prepared using gallic acid as the standard: Abs = (6.7 ± 0.6) × 10^−3^ (L/mg) C (mg/L) + (2.3 ± 1.9) × 10^−4^; correlation coefficient (R^2^) = 0.997; limit of detection (LOD) = 0.5 mg/L; limit of quantification (LOQ) = 1.7 mg/L; linear range: 0.5 mg/L–20.0 mg/L. Each analysis was performed in triplicate. The results were expressed as mg of gallic acid equivalents (mg_GAE_) per g of dry weight (dw) of OMP (mg_GAE_/g_OMP_).

The scavenging activity of extracts was extrapolated in the inhibitory percentage of DPPH^•^ (% I) Equation (1):(1)$$\%\;I=\frac{A_{0}-A_{1}}{A_{0}}\times 100$$
where, $A_{0}$ is the absorbance of the control (DPPH radical in methanol) and $A_{1}$ is the absorbance of the DPPH radical plus tested sample in methanol.

The calibration curve obtained was based on the mean of three independent calibration curves: Abs = (3.9 ± 0.1) × 10^−3^ C (mg/L) + (3.8 ± 0.3) × 10^−2^; R^2^ = 0.998; LOD = 3.0 mg/L; LOQ = 10.0 mg/L; linear range: 3.1 mg/L–200.0 mg/L.

#### 3.2.2. Analysis of Individual Phenolic Compounds by Reserved-Phase High-Performance Liquid Chromatography (RP-HPLC)

The qualitative and quantitative assessment of hydroxytyrosol, tyrosol, and oleuropein in the OMP extracts obtained under optimized extraction conditions, was performed by RP-HPLC analysis using gradient elution. Analyses were performed using a Merck Hitachi Elite LaChrom (Tokyo, Japan) high-performance liquid chromatograph equipped with a Hitachi L-7100 pump and L-7250 autosampler and coupled to a L-7450A diode array detector. Samples and standards were injected into the Purospher^®^ STAR RP-18 end-capped LiChroCART^®^ column (250 mm × 4.0 mm, 5.0 μm) (Merck KGaA), attached to a guard column (4.0 mm × 4.0 mm, 5.0 μm) of the same type. The elution method applied was adapted from Tasioula-Margari and Tsabolatidou [35], and was carried out at a flow rate of 1 mL/min, considering the following two-buffer gradient system: (A) UPW/acetic acid (0.1 N) (97.5/2.5 *v*/*v*); (B) acetonitrile/methanol (50/50 *v*/*v*). The following gradient elution was applied: 95% A and 5% B as initial conditions that were kept for 5 min, 70% A and 30% B during 30 min, and then re-establishing the initial conditions (95% A and 5% B) for 5 min (from minute 35 to minute 40)). The absorbance of the eluate at 280 nm was recorded at 1 s intervals. The linearity was evaluated by the direct injection of calibration standards prepared in UPW, which contained the target phenolic antioxidants. The results were expressed as mg of each phenolic antioxidant per g of dry weight (dw) of OMP (mg/g dw OMP). The amount of each specific phenolic antioxidant (hydroxytyrosol, tyrosol, and oleuropein) recovered after the optimized extraction procedure was evaluated through the calculation of the recovery (%R) percentage considering the standard addition method as presented in Equation (2):(2)$$\%R=\frac{m_{\mathrm{is}}-m_{\mathrm{ius}}}{m_{\mathrm{ia}}}\times 100$$
where, $m_{\mathrm{is}}$ is the mass of the specific antioxidant i in the spiked sample, $m_{\mathrm{ius}}$ is the mass of the antioxidant i in the unspiked sample and $m_{\mathrm{ia}}$ is the added mass of the antioxidant i to the sample.

### 3.3. Design of Experiments for the Optimization of the Extraction Procedure

#### 3.3.1. Experimental Design

In this work, a central composite rotatable response surface design (CCRD) was employed as a mathematical and statistical design to optimize the effect of selected process variables such as (i) X_1_—the time of the primary extraction (hours), (ii) X_2_—the SSR (mL/g) during the primary extraction step, (iii) X_3_—the time of the secondary extraction step (hours) and (iv) X_4_ –the SSR during the second extraction (mL/g) on the response Y—total phenolic content (mg GAE/g dw OMP). The four process variables were coded in five levels (−2,−1,0,1,2).

The coding of the variables was performed as follow Equation (3) [17,36]:(3)$$x_{i}=\frac{X_{i}-X_{z}}{\Delta X_{i}},\;i=1,2,\ldots ,k$$
where, x_i_ corresponds to the dimensionless coded value of an independent variable, $X_{i}$ the actual value of the independent variable i, $X_{z}$ is the real value of the independent variable at the center point and $\Delta X_{i}$ corresponds to the step change of the real value of the variable under study, i. The CCRD consisted of 30 experiments, containing 16 factorial points, 8 axial points, 6 center points considering one block run.

The total number of experiments, N, was evaluated considering Equation (4) [37].
(4)$$N=2^{K}+2K+C_{p}$$
where K is the number of process variables (in the present study, K=4), $2^{K}$ corresponds to the number of factorial points in the design ($2^{K}=2^{4}=16)$, 2K the number of axial points on the axis of each design factor at a distance of ±α (for $K=4,\;\alpha =2^{K/4}=2,\;2K=8$) and $C_{p}$ is the number of replicates at the center point ($C_{p}=6)$. The design was randomized in order to reduce the errors from the experimental process relative to extraneous factors. The experimental data were fitted in a second-order polynomial Equation (5) using a non-linear regression method, expressing, therefore, the mathematical relationship between the process variables (from X_1_ to X_4_) and the responses Y.
(5)$$Y=\beta_{0}+\sum_{j=1}^{K}\beta_{j}x_{j}+\sum_{j=1}^{K}\beta_{\mathrm{jj}}x_{j}^{2}+\sum_{i}\sum_{<j=2}^{K}\beta_{\mathrm{ij}}x_{i}x_{j}+e_{i}$$
where Y corresponds to the response, $\beta_{0}$ is the model intercept coefficient, $\beta_{j}$, $\beta_{\mathrm{jj}}$ and $\beta_{\mathrm{ij}}$ are the interaction coefficients of the linear, quadratic and second-order terms, respectively, K is the number of independent variables (K=4) and $e_{i}$ is the error.

#### 3.3.2. Statistical Analysis

The statistical software Design Expert 12 (Stat-Ease Inc, MN, USA) was employed for the analysis of the obtained experimental data. Experimental data were evaluated using multiple regressions analysis and the Pareto analysis of variance. The linear, quadratic, and interaction terms for the responses were found through analysis of variance (ANOVA), and the significance of terms was evaluated considering the F-statistical value obtained. Descriptive statistical analysis including the calculation of (i) the p-value, (ii) the F-value, (iii) the degrees of freedom (DF), (iv) the sum of squares, (v) the coefficient of variation, (vi) the determination coefficient ($R^{2}$), (vii) the adjusted determination coefficient ($R_{a}^{2}$) and the (viii) correlation coefficient (R) were calculated in order to evaluate the statistical significance of the model. Experimental data were fitted into a quadratic model that was used to sketch the three-dimensional surface plots.

#### 3.3.3. Contributions of Process Variables

The percentage contribution (PC) of each process variables was assessed based on the regression coefficients from the ANOVA and evaluated as described by Khataee, et al. [38] and Prakash Maran, Manikandan, Thirugnanasambandham, Vigna Nivetha and Dinesh [36]
(6)$${\mathrm{PC}}_{i}=\left(\frac{\beta_{i}^{2}}{\sum \beta_{i}^{2}}\right)\times 100\;\;\;\left(i\neq 0\right)$$
where $\beta_{i}$ corresponds to the regression coefficient of the individual process variable.

#### 3.3.4. Determination of the Optimized Extraction Conditions

The Derringer’s desired function methodology was considered to assess the optimal extraction conditions to maximize simultaneously the responses of the study [39]. The generic assessment of the desirability function is based firstly on the transformation of a specific response into a dimensionless individual desirability function ($d_{i}$) that ranges between 0 (lowest desirability) to 1 (highest desirability). The overall desirability function (G) was obtained considering the geometric means of each $d_{i}$ as follows Equation (7):(7)$$G={\left(d_{1}\times d_{2}\times d_{3}\times \ldots \times d_{n}\right)}^{1/n}$$
where $d_{i}$ denotes the desirability of the response i and n represent the number of responses of the study. If any of the responses are beyond the desirability, the G turns into zero.

The G can be extended, considering the importance of the responses Equation (8):(8)$$G={\left(d_{1}^{\alpha 1}\times d_{2}^{\alpha 2}\times d_{3}^{\alpha 3}\times \ldots \times d_{n}^{\frac{1}{n}}\right)}^{\frac{1}{n}}\;,\;0\leq \alpha_{i}\leq 1\;\left(i=1,2,3,\ldots ,n\right),{\;\alpha }_{1}+\alpha_{2}+{\;\alpha }_{3}+\ldots +\alpha_{n}=1$$
where $\alpha_{i}$ represents the importance of the response $Y_{i}\;\left(i=1,2,3,\ldots ,n\right)$ [40]. According to this optimization approach, response goals should be assigned to a low (LV) and a high value (HV). In the present study, a desirability function criterion Equation (9) was employed to maximize the response.
(9)$$\begin{array}{l} d_{i}=0\;\mathrm{if}\;Y_{i}<\mathrm{LV}\\ 0\leq d_{i}\leq 1\;\mathrm{if}\;\mathrm{LV}<Y_{i}<\mathrm{HV}\\ d_{i}=0\;\mathrm{if}\;Y_{i}>\mathrm{HV} \end{array}$$

The shape of the specific desirability response can be modeled through the weight factor (WF). A WF of 1 creates a linear function between the HV and the LV and the goal or between the HV and the goal. Increasing the WF (up to 10) switch the result towards the goal and reducing the WF (down to 0.1) generates the opposite effect [36]. In the present study, a default WF of 1 was chosen. The default importance of 3 was chosen, representing that the goals should be considered of equal importance.

A second-order polynomial equation was generated, applying multiple regression analyses on the experimental data. The final Equation obtained in terms of uncoded variables (real values) is presented in the following Equation (10):(10)$$\mathrm{TPC}\;=100.9+27.7{\;X}_{1}+7.2{\;X}_{2}-57.3{\;X}_{3}-43.9{\;X}_{4}-0.1{\;X}_{1}X_{2}+0.9{\;X}_{1}X_{3}-2.1{\;X}_{1}X_{4}+\;5.4{\;X}_{2}X_{3}-2.5{\;X}_{2}X_{4}+12.9{\;X}_{3}X_{4}-3.5{\;X}_{1}^{2}-0.3{\;X}_{2}^{2}-14.9{\;X}_{3}^{2}+6.2{\;X}_{4}^{2}$$
where TPC is the total phenolic content (expressed in mg GAE/g dw OMP) and $X_{1}$, $X_{2}$, $X_{3}$ and $X_{4}$ are the uncoded variables of the time of the primary extraction (hours), the SSR during the primary extraction (mL/g), the time of the secondary extraction (hours), and the SSR during the secondary extraction (mL/g), respectively.

#### 3.3.5. Verification of the Predicted Optimized Extraction Conditions

The applicability and accuracy of the generated model were assessed through triplicate experiments under the optimized conditions predicted by the model. The adequacy of the optimized model was evaluated by comparing the average values obtained for the response at optimized conditions and the predicted values by the model.

## 4. Conclusions

In this study, a central composite rotatable response surface design was employed to a two-step solid-liquid extraction to investigate and optimize the influence of the individual and the interactive effect of process variables as the time of the primary extraction, the solvent-to-sample ratio during the primary extraction, the time of the secondary extraction and the solvent-to-sample ratio during the secondary extraction on the total phenolic content of olive mill pomace extracts. The results suggested that the process variables had a significant effect on the recovery of phenolic antioxidants estimated through the total phenolic content of olive mill pomace extracts. The quadratic model F-value and the associated *p*-values confirmed the significance of the selection of the quadratic regression model. The greater part of the antioxidants extracted were phenolic compounds, and most phenolic compounds were hydrophilic in nature. The acidic hydrolysis presented the utmost importance for the extraction of phenolic antioxidants, as this step allowed the efficient extraction of phenolic antioxidants linked to olive mill pomace cells by ester and glycosidic bonds and the dissolution of phenolic antioxidants retained inside of olive mill pomace cells. The Derringer’s desired function methodology allowed for optimization the extraction conditions, leading to a predicted maximum total phenolic content of olive mill pomace extracts. This methodology could be applied to ensure that polyphenol extracts were efficiently obtained from by-products, such as the olive stone in the food industry, allowing for a readily scalable addition of a source of income to olive farmers and olive oil processors.

## Figures and Tables

**Figure 1 molecules-27-08620-f001:**
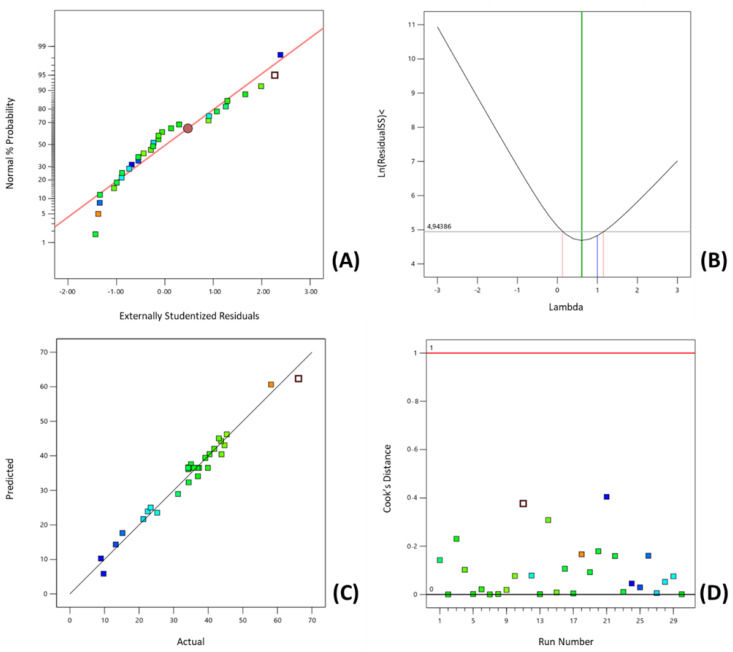
Diagnostic plots for the quadratic model adequacy to the total phenolic content of olive mil pomace extracts ((**A**)—Normal probability plot; (**B**)—Box-Cox plot for power transforms; (**C**)—Predicted versus Actual plot; (**D**)—Cook’s Distance plot).

**Figure 2 molecules-27-08620-f002:**
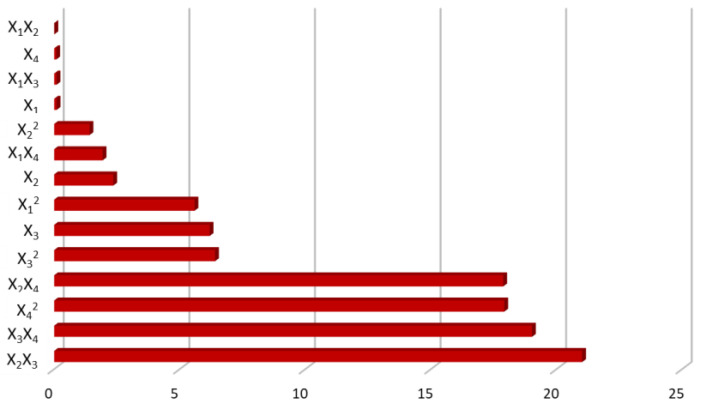
Detailed diagram of the percentage contribution of the linear, the two-factor interaction and the quadratic effects on the total phenolic content of olive mill pomace extracts.

**Figure 3 molecules-27-08620-f003:**
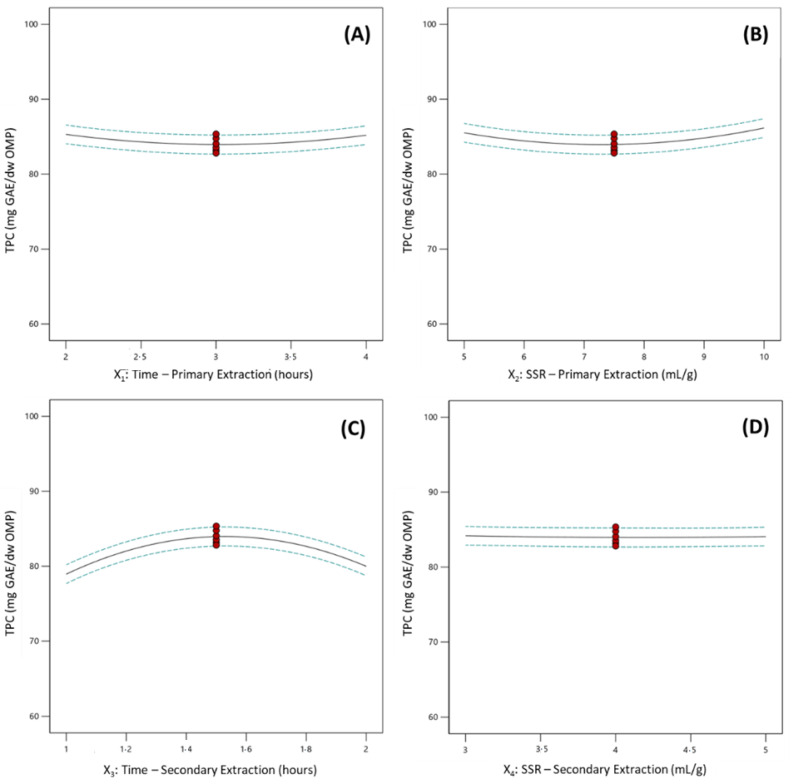
One variable plots portraying the effect of each process variable on the total phenolic content of the extracts obtained from olive mill pomace samples ((**A**)—effect of the time of the primary extraction; (**B**)—effect of the solvent-to-sample ratio during the primary extraction; (**C**)—effect of the time of the secondary extraction; (**D**)—effect of the solvent-to-sample ratio during the secondary extraction; 
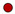
 represent design points; 
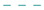
 represent 95% confident interval band).

**Figure 4 molecules-27-08620-f004:**
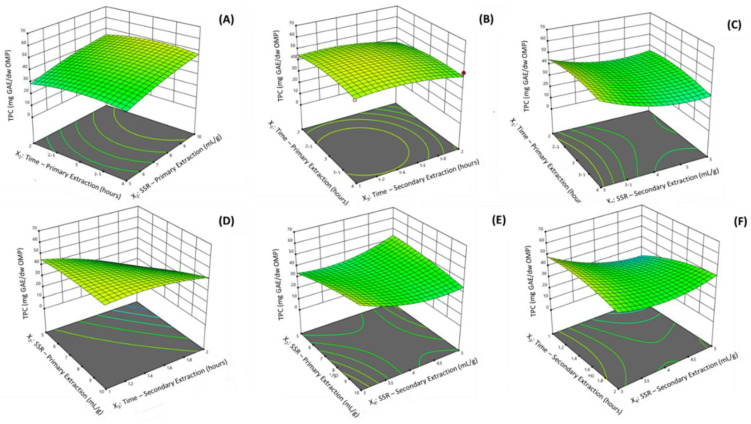
Response surface plots representing the effect of each process variable on the total phenolic content of the extracts obtained from OMP samples (
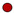
 represent a design point). (**A**) X_1_: Time—Primary Extraction (hours) versus X_2_: solvent-to-sample ratio (SSR)—Primary Extraction (mL/g); (**B**) X_1_ versus X_3_: Time—Secondary Extraction (hours); (**C**) X_1_ versus X_4_: Time—Secondary Extraction (hours); (**D**) X_2_ versus X_3_; (**E**) X_3_ versus X_4_; (**F**) X_3_ versus X_4_.

**Figure 5 molecules-27-08620-f005:**
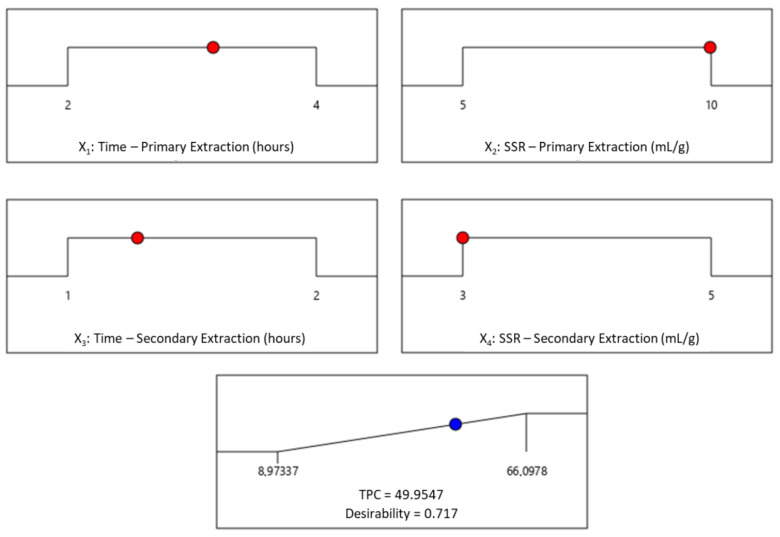
Desirability ramp for the optimization of the two-step extraction of phenolic antioxidants from olive mill pomace (
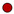
 represent design points; 
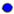
 represent a predicted point).

**Figure 6 molecules-27-08620-f006:**
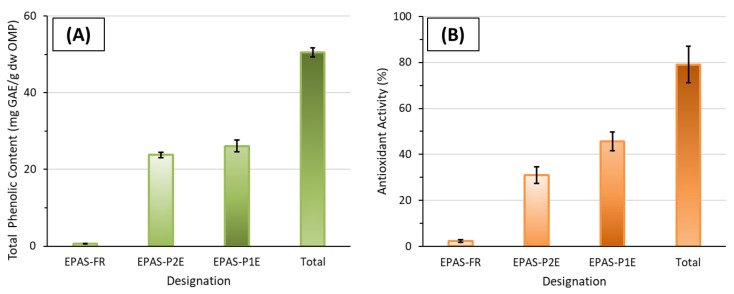
Distribution of the total phenolic content (**A**) and the total antioxidant activity (**B**) among different fractions (EPAS-FR, EPAS-P2E, and EPAS-P1E) of OMP extracted samples obtained considering extraction optimized conditions (EPAS-FR—Extracted Phenolic Antioxidants during the Fat Removal procedure; EPAS-P1E—Extracted Phenolic Antioxidants during the primary extraction; EPAS-P2E—Extracted Phenolic Antioxidants from the secondary extraction procedure).

**Table 1 molecules-27-08620-t001:** Experimental design for the optimization of the extraction of antioxidants and phenolic compounds from olive mill pomace.

Run No.	Process Variables (Coded Variables)				Response		
	X_1_: Time P1E (hours) ^a^	X_2_: P1E-SSR (mL/g) ^a^	X_3_: Time P2E (hours) ^a^	X_4_: P2E-SSR (mL/g) ^a^	Y: TPC (mg GAE/g dw OMP)		
					OV	PV	RE
1	3.0 (0)	7.5 (0)	0.5 (−2)	4.0 (0)	31.3	29.0	2.3
2	3.0 (0)	7.5 (0)	1.5 (0)	4.0 (0)	36.9	36.6	0.4
3	3.0 (0)	12.5 (2)	1.5 (0)	4.0 (0)	37.0	34.1	2.9
4	2.0 (−1)	5.0 (−1)	1.0 (−1)	5.0 (1)	43.1	45.0	−2.0
5	4.0 (1)	5.0 (−1)	1.0 (−1)	5.0 (1)	39.1	39.4	−0.3
6	3.0 (0)	7.5 (0)	1.5 (0)	4.0 (0)	39.9	36.6	3.3
7	2.0 (−1)	5.0 (−1)	1.0 (−1)	3.0 (−1)	40.4	40.5	−0.1
8	2.0 (−1)	10.0 (1)	2.0 (1)	5.0 (1)	41.8	4.0	−0.3
9	4.0 (1)	10.0 (1)	1.0 (−1)	3.0 (−1)	45.4	46.2	−0.8
10	4.0 (1)	5.0 (−1)	1.0 (−1)	3.0 (−1)	44.7	43.0	1.7
11	3.0 (0)	7.5 (0)	1.5 (0)	6.0 (2)	66.1	62.4	3.7
12	1.0 (−2)	7.5 (0)	1.5 (0)	4.0 (0)	25.3	23.5	1.7
13	3.0 (0)	7.5 (0)	1.5 (0)	4.0 (0)	37.4	36.6	0.8
14	4.0 (1)	10.0 (1)	2.0 (1)	3.0 (−1)	43.8	40.4	3.4
15	2.0 (−1)	10.0 (1)	1.0 (−1)	3.0 (−1)	43.7	44.3	−0.6
16	4.0 (1)	5.0 (−1)	2.0 (1)	5.0 (1)	34.3	32.3	2.0
17	3.0 (0)	7.5 (0)	1.5 (0)	4.0 (0)	35.1	36.6	−1.5
18	3.0 (0)	7.5 (0)	1.5 (0)	2.0 (−2)	58.1	60.6	−2.5
19	2.0 (−1)	5.0 (−1)	2.0 (1)	5.0 (1)	34.3	36.1	−1.9
20	4.0 (1)	10.0 (1)	2.0 (1)	5.0 (1)	35.0	37.6	−2.6
21	2.0 (−1)	5.0 (−1)	2.0 (1)	3.0 (−1)	9.7	5.9	3.9
22	2.0 (−1)	10.0 (1)	2.0 (1)	3.0 (−1)	34.2	36.7	−2.4
23	3.0 (0)	7.5 (0)	1.5 (0)	4.0 (0)	34.2	36.6	−2.3
24	4.0 (1)	5.0 (−1)	2.0 (1)	3.0 (−1)	9.0	10.3	−1.3
25	3.0 (0)	7.5 (0)	2.5 (2)	4.0 (0)	13.3	14.3	−1.0
26	4.0 (1)	10.0 (1)	1.0 (−1)	5.0 (1)	15.2	17.7	−2.4
27	5.0 (2)	7.5 (0)	1.5 (0)	4.0 (0)	21.2	21.7	−0.5
28	2.0 (−1)	10.0 (1)	1.0 (−1)	5.0 (1)	22.5	23.9	−1.4
29	3.0 (0)	2.5 (−2)	1.5 (0)	4.0 (0)	23.4	25.0	−1.7
30	3.0 (0)	7.5 (0)	1.5 (0)	4.0 (0)	35.9	36.6	−0.7

^a^ The values presented between brackets represent the coded values for the variables of the study. %E—Percentage of Error; dw—dry weight; GAE—Gallic Acid Equivalents; OMP—Olive Mill Pomace; OV—Observed Value; P1E—Primary Extraction; P2E—Secondary Extraction; PV—Predicted Value; RE—Residual Error; SSR—Solvent-to-Sample Ratio; TPC—Total Phenolic Content.

**Table 2 molecules-27-08620-t002:** Sequential model fitting for the extraction of phenolic antioxidants from olive mill pomace extracts.

Y: TPC (mg GAE/g dw OMP)	**Model**	**Source**	**Sum of Squares**	**DF**	**Mean Square**	**F-Value**	***p*-Value**	**Remarks**
	Sequential model sum of squares	Mean	3544.7	1	35,447.7	-	-	-
		Linear	454.5	4	113.6	0.65	0.6300	-
		2FI	2081.8	6	347.0	2.90	0.0400	-
		Quadratic	2151.2	4	537.8	64.51	<0.0001	Suggested
		Cubic	83.3	8	10.4	1.75	0.2400	Aliased
		Residual	41.8	7	6.0	-	-	-
		Total	40,260.2	30	1342.0	-	-	-
	**Model**	**Source**	**Sum of Squares**	**DF**	**Mean Square**	**F-Value**	***p*-Value**	**Remarks**
	Lack-of-fit Tests	Linear	4338.0	20	216.9	54.27	0.0002	-
		2FI	2256.3	14	161.1	40.33	0.0003	-
		Quadratic	105.1	10	10.5	2.63	0.1488	Suggested
		Cubic	21.8	2	10.9	2.73	0.1583	Aliased
		Pure Error	20.0	5	4.0	-	-	-
	**Model**	**Source**	**Std. dev.**	**R^2^**	**Adjusted R^2^**	**Predicted R^2^**	**PRESS**	**Remarks**
	Summary Statistics	Linear	13.2	0.094	−0.050	−0.410	6785.6	-
		2FI	11.0	0.527	0.278	0.205	3827.5	-
		Quadratic	2.9	0.074	0.950	0.868	634.0	Suggested
		Cubic	2.4	0.991	0.964	0.342	3165.4	Aliased
	**Fit Summary**	**Source**	**Sequential *p*-Value**		**Lack of fit *p*-Value**	**Adjusted R^2^**	**Predicted R^2^**	**Remarks**
		Linear	0.6310		0.0002	−0.054	−0.410	-
		2FI	0.0354		0.0003	0.278	0.201	-
		Quadratic	<0.0001		0.1488	0.950	0.868	Suggested
		Cubic	0.2387		0.1583	0.964	0.342	Aliased

2FI—Two Factors Interaction; DF—Degrees of Freedom; dw—dry weight; OMP—Olive Mill Pomace; PRESS—Predicted Residual Error Sum of Squares; R^2^—Determination Coefficient; Std. Dev.—Standard Deviation; TPC—Total Phenolic Content.

**Table 3 molecules-27-08620-t003:** Analysis of variance for the total phenolic content of olive mill pomace extracts.

**Y: TPC (mg GAE/g dw OMP)**	**Source**	**Coefficient Estimate**	**Standard Error**	**Sum of Squares**	**DF**	**Mean Square**	**F-Value**	***p*-Value**	**Remarks**
	Model	-	-	4687.5	14	334.8	40.16	<0.0001	S
	Intercept	36.6	1.2	-	1	-	-	-	-
	X_1_	−0.5	0.6	5.1	1	5.1	0.62	0.4459	-
	X_2_	2.3	0.6	122.7	1	122.7	14.71	0.0016	-
	X_3_	−3.7	0.6	322.3	1	322.3	38.66	<0.0001	-
	X_4_	0.4	0.6	4.4	1	4.4	0.53	0.4781	-
	X_1_X_2_	−0.2	0.7	0.4	1	0.4	0.05	0.8313	-
	X_1_X_3_	0.5	0.7	3.4	1	3.4	0.41	0.5313	-
	X_1_X_4_	−2.1	0.7	67.0	1	67.0	8.03	0.0126	-
	X_2_X_3_	6.8	0.7	729.8	1	729.8	87.54	<0.0001	-
	X_2_X_4_	−6.2	0.7	620.7	1	620.7	74.45	<0.0001	-
	X_3_X_4_	6.4	0.7	660.5	1	660.5	79.22	<0.0001	-
	X_1_^2^	−3.5	0.6	332.7	1	332.7	39.90	<0.0001	-
	X_2_^2^	−1.6	0.6	83.8	1	83.8	10.05	0.0063	-
	X_3_^2^	−3.7	0.6	381.3	1	381.3	45.73	<0.0001	-
	X_4_^2^	6.2	0.6	1066.5	1	1066.5	127.93	<0.0001	-
	Residual	-	-	125.06	15	8.34	-	-	-
	Lack of Fit	-	-	105.07	10	10.51	2.63	0.1488	NS
	Pure Error	-	-	19.98	5	4.00	-	-	-
	Cor Error	-	-	4812.52	29	-	-	-	-
	**Std. Dev.**	**Mean**	**CV (%)**	**R^2^**	**Adjusted R^2^**	**Predicted R^2^**	**Adeq. precision**	**PRESS**	-
	2.9	34.4	8.4	0.974	0.950	0.868	27.7	634.0	-

Abbreviations: CV (%)–Coefficient of Variance; DF—Degrees of Freedom; dw—dry weight; GAE—Gallic Acid Equivalents; NS—Not Significant; OMP—Olive Mill Pomace; PRESS—Predicted Residual Error Sum of Squares; R^2^—Determination Coefficient; S—Significant; Std. Dev.—Standard Deviation; TPC—Total Phenolic Content; X_1_—Coded variable for the time of the primary extraction (hours); X_2_—Coded variable for the sample-to-solvent during the primary extraction (mL/g); X_3_—Coded variable for the time of the secondary extraction (hours); X_4_—Coded variable for the sample-to-solvent during the secondary extraction (mL/g).

**Table 4 molecules-27-08620-t004:** Experimental and Post analysis of the response.

Post Analysis	
Predicted (mg GAE/g of dw OMP)	50.0
95% PI low (mg GAE/g of dw OMP)	45.2
95% PI high (mg GAE/g of dw OMP)	54.7
Experimental (mg GAE/g of dw OMP) ^a^	50.5 ± 1.5
Residual Error	0.5
%Error	1.0

^a^ Data expressed as mean ± standard deviation of three independent extractions considering the optimized extraction conditions. dw—dry weight; GAE—Gallic Acid Equivalents; OMP—Olive Mill Pomace; PI—Predicted Interval.

**Table 5 molecules-27-08620-t005:** Total phenolic content and their proportion of different factions of olive mill pomace extracted full-fat samples and extracted defatted samples.

Extract Fraction	Hydrolyzed Extracted Samples				Non-Hydrolyzed Extracted Samples			
	TPC (mg GAE/g dw OMP)	TPC Proportion (%)	TAA (%)	TAA Proportion (%)	TPC (mg GAE/g dw OMP)	TPC Proportion (%)	TAA (%)	TAA Proportion (%)
**TPA**	50.5 ± 1.5	100	79.1 ± 7.9	100	21.8 ± 0.7	100	53.5 ± 3.9	100
**BPA**	13.6 ± 1.1	26.9 ± 2.2	29.1 ± 3.9	36.8 ± 4.9	2.0 ± 0.1	9.2 ± 0.5	18.8 ± 4.6	35.1 ± 7.3
**FPA**	36.9 ± 0.4	73.1 ± 0.8	49.9 ± 4.7	63.1 ± 5.9	19.8 ± 0.6	91.2 ± 2.8	34.7 ± 3.8	64.9 ± 7.1

BPA—Bounded Phenolic Antioxidants; dw—dry weight; FPA—Free Phenolic Antioxidants; GAE—Gallic Acid Equivalents; OMP—Olive Mill Pomace; TAA—Total Antioxidant Activity; TPA—Total Phenolic Antioxidants; TPC—Total Phenolic Content.

## Data Availability

The data presented in this study are available on request from the corresponding authors.

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
