# Peer review of "Response Surface Modeling and Optimization of the Extraction of Phenolic Antioxidants from Olive Mill Pomace"

_molecules, 2022, doi:10.3390/molecules27238620_

Round 1

Reviewer 1 Report

The study employed different statistical techniques to demonstrate the optimal combination of processes to enhance the yield of phenolics etc.

Authors please address the following:

Section 3.1; 3.2.1; 3.2.2 What quality control procedure, i.e. quality control samples were included to maintain the accuracy?

Author Response

Reviewer #1:

Comment: The study employed different statistical techniques to demonstrate the optimal combination of processes to enhance the yield of phenolics, etc. Authors please address the following: Section 3.1; 3.2.1; 3.2.2 What quality control procedure, i.e. quality control samples were included to maintain the accuracy?

Response: Thank you for the salient comment. The OMP samples were stored at - 22 °C before analysis. The samples were freeze-dried for 72 hours on a benchtop freeze-dryer (SP Scientific, NY, USA) (moisture content of 71.0 ± 5.5 % w/w) and grounded on an electric mill (Qilive Q5321 Grinder) to an average particle size of 142.2 ± 9.6 μm. The mean particle size of OMP was evaluated by laser granulometry technique using a Coulter Counter-LS 230 Particle Size Analyzer (Miami, FL, USA) equipment. These procedures were applied in order to optimizing sample preparation procedures for accurate estimation of phenolic compounds. In addition, in the analysis of individual phenolic compounds by reserved-phase high-performance liquid chromatography (RP-HPLC), for each phenolic antioxidant, the calibration curve was obtained based on the mean of three independent calibration curves. In the case of hydroxtyrosol: Abs = (5.5 ± 0.2) x 105 C (mg/L) + (4.6 ± 0.8) x 105; R = 0.997; LOD = 0.7 mg/L; LOQ = 2.2 mg/L; linear range: 5.0 mg/L – 20.0 mg/L. In the case of tyrosol: Abs = (8.4 ± 0.7) x 105 C (mg/L) + (-6.8 ± 0.3) x 103; R = 0.995; LOD = 1.0 mg/L; LOQ = 3.2 mg/L; linear range: 5.0 mg/L – 20.0 mg/L. The information was added to the revised manuscript (Lines 568, 625 and 637).

Reviewer 2 Report

The manuscript entitled “Response surface modeling and optimization of the extraction of phenolic antioxidants from olive mill pomace” is well presented and the obtained results could be of relevance for the scientific community. I will recommend it for publishing in the Molecules journal after some corrections based on comments I made below.

 Introduce abbreviation GAE in the abstract.

Table 1 should be placed within Section 2 or to be renumered. Also Table 5 should be within Section 2.7.3

Line 212. Correct the section number.

Line 214. there is no section 2.4.3

line 358. Is „n“ refer to total number of extracts?

line 373. The names of these abbreviation should be introduced here rather than under Figure 6. I get confused initially during reading this section.

How you determined the concentration of bounded and free phenolics?

383-386 What means „from hydrophilic nature“?

Lines 563 and 579 Supplementary material is missing

Author Response

Reviewer #2

Comment: The manuscript entitled “Response surface modeling and optimization of the extraction of phenolic antioxidants from olive mill pomace” is well presented and the obtained results could be of relevance for the scientific community. I will recommend it for publishing in the Molecules journal after some corrections based on comments I made below.

Response: The authors would like to thank Reviewer 2 for taking into account the scientific quality of our manuscript and considering it worth of publication after minor revision.

Comment: Introduce abbreviation GAE in the abstract.

Response: Thank for the observation. The abbreviation GAE was introduced in the abstract of the revised manuscript (Line 25).

Comment: Table 1 should be placed within Section 2 or to be renumered. Also Table 5 should be within Section 2.7.3

Response: Thank for the observation. Table 1 was placed within Section 2 and Table 5 in the section Section 2.7.3. Please see the revised manuscript (Line 102 and 551).

Comment: Line 212. Correct the section number.

Response: Section number was corrected in the revised manuscript (226).

Comment: Line 214. there is no section 2.4.3

Response: The reference to the section 2.4.3 was removed.

Comment: line 358. Is „n“ refer to total number of extracts?

Response: Thank you for the observation. Yes, “n” refer to total number of extracts. This information was added to the revised manuscript (Line 372).

Comment: line 373. The names of these abbreviation should be introduced here rather than under Figure 6. I get confused initially during reading this section.

Response: Thanks for the observation. The manuscript was corrected accordingly (Line 392).

Comment: How you determined the concentration of bounded and free phenolics?

Response: Thank you for the salient observation. The extraction of bioactive compounds were performed in three stages: (i) sample pre-treatment and bonded phenolic compounds extraction, (ii) two-step solid-liquid extraction of free phenolic compounds, and (iii) liquid-liquid extraction of these valuable bioactive compounds, as presented in the following sections. The (i) acidic hydrolysis promotes the breakage of both glycosidic and ester bonds [17], and (ii) the fat removal procedure was considered, as lipids, due to their lipophilic nature, are interfering agents in obtaining hydroalcoholic extracts [7]. The information was added to the revised manuscript (Lines 573 and 580)

Comment: 383-386 What means „from hydrophilic nature“? 

Response: Thanks for the observation. Hydrophilic nature is referring the aromatic rings with attached hydroxyl groups in their structures with strong polar character). This information was added (408).

Comment: Lines 563 and 579 Supplementary material is missing

Response: The supplementary information was included (Please, see the file of Supplementary material).

Reviewer 3 Report

The paper seeks to optimize the extraction process of polyphenols from olive mill pomace by measuring different parameters and the topic is critical in the field. However, the manuscript needs to be revised prior to further consideration as it has some minor problems. The references are not up-to-date and the authors should discuss some more updated papers. Moreover, the manuscript needs to be checked by a native English speaker because there are many spelling and grammatical mistakes.

Abstract: Please rewrite the last sentence of the abstract as it is vague. Write a new sentence showing the conclusion of the paper and putting the paper in the scope. 

Line 23: Please modify the unit of TPC (mg GAE/ g (DW or FW???)). You should specify if it is on the basis of DRY WEIGHT (DW) or FRESH WEIGHT (FW). (Also, in the other parts of the paper)

Keywords: The keywords could not be the same as the words in the title, please consider changing them.

Introduction:

Line 60-62: There are many papers studying the optimization of extraction processes using traditional methods (solvent extraction) from different materials including olive mill pomace. Please show the novelty of your work by focusing on the other aspects of your research and delete this sentence.

You can write the adverse effects of traditional extraction methods and conclude that by optimization you can decrease these adverse effects. It is necessary to mention that traditional methods are not reliable anymore as there are many more Environmentally Friendly Techniques for the Recovery of Polyphenols from Food By-Products.

Results and discussions:

The numbering of tables is not correct. It starts from table 2. Please solve this problem.

In my opinion, the inclusion of table 2 in the paper is not necessary and it is better to report the used model in the text, instead.

The paper lacks enough discussion about the previous studies on the topic. Please include the most relevant research conducted in the field.

Please define Abbreviations only in the first time they appear in each section including the abstract; the main text; the first figure or table.

Materials and methods:

Line 586: Please write a sentence reporting that a standard curve of gallic acid was used including the equation of the curve, R2, and linearity range.

Conclusion:

The conclusion is too long and broad. It is not correct to report everything again in the conclusion. Please re-write the conclusion by summarizing the most important findings of the paper and including the future trends of the topic.

Author Response

Reviewer #3

Comment:   The paper seeks to optimize the extraction process of polyphenols from olive mill pomace by measuring different parameters and the topic is critical in the field. However, the manuscript needs to be revised prior to further consideration as it has some minor problems. The references are not up-to-date and the authors should discuss some more updated papers. Moreover, the manuscript needs to be checked by a native English speaker because there are many spelling and grammatical mistakes.

Response: The authors would like to thank Reviewer 3 for taking into account the scientific quality of our manuscript and considering it worth of publication after minor revision. Following your feedback, which we truly appreciate, our team (which includes a native English speaker) revised the article to eliminate any misinterpretations.

Comment:   Please rewrite the last sentence of the abstract as it is vague. Write a new sentence showing the conclusion of the paper and putting the paper in the scope.

Response: Thanks for the observation. The abstract section has been rewritten accordingly (Line 24).

Comment:   Line 23: Please modify the unit of TPC (mg GAE/ g (DW or FW???)). You should specify if it is on the basis of DRY WEIGHT (DW) or FRESH WEIGHT (FW). (Also, in the other parts of the paper)

Response: Thanks for the observation. The unit of TPC was in DRY WEIGHT (DW). The information was included in the revised manuscript (Line 25).

Comment:   Keywords: The keywords could not be the same as the words in the title, please consider changing them.

Response: Thanks for the observation. The keywords were changed accordingly (Line 29).

Comment:   Line 60-62: There are many papers studying the optimization of extraction processes using traditional methods (solvent extraction) from different materials including olive mill pomace. Please show the novelty of your work by focusing on the other aspects of your research and delete this sentence.

Response: Thanks for the observation. The information was included (Line 85).

Comment:   You can write the adverse effects of traditional extraction methods and conclude that by optimization you can decrease these adverse effects. It is necessary to mention that traditional methods are not reliable anymore as there are many more Environmentally Friendly Techniques for the Recovery of Polyphenols from Food By-Products.

Response: Thanks for the observation. The information was included (Line 60).

Comment:   The numbering of tables is not correct. It starts from table 2. Please solve this problem.

Response: We Table numbers were changed in the revised manuscript Please see the revised manuscript (Line 102 and 551).

Comment:   In my opinion, the inclusion of table 2 in the paper is not necessary and it is better to report the used model in the text, instead.

Response: Thanks for the salient suggestion. However, we opted to maintain the models in the Table 2 in order to facilitate the reading of the Journal readers, having all the models in the same place, which facilitate the comparison between them.

Comment:   The paper lacks enough discussion about the previous studies on the topic. Please include the most relevant research conducted in the field.

Response: Thanks for the observation.  The most relevant research are highlighted in blue color in the revised manuscript.

Comment:   Please define Abbreviations only in the first time they appear in each section including the abstract; the main text; the first figure or table.

Response: Thank you. The manuscript was revised accordingly (see section of abstract, results and materials and methods).

Comment:  Materials and methods: Line 586: Please write a sentence reporting that a standard curve of gallic acid was used including the equation of the curve, R2, and linearity range.

Response: The information was included (Line 625 and 637).

Comment: Conclusion: The conclusion is too long and broad. It is not correct to report everything again in the conclusion. Please re-write the conclusion by summarizing the most important findings of the paper and including the future trends of the topic.

Response: The section of conclusion was changed, so as to be short, and the most important findings of the paper and including the future trends of the topic were included (Line 774).
